# Focus on Nitric Oxide Homeostasis: Direct and Indirect Enzymatic Regulation of Protein Denitrosation Reactions in Plants

**DOI:** 10.3390/antiox11071411

**Published:** 2022-07-21

**Authors:** Patrick Treffon, Elizabeth Vierling

**Affiliations:** Department of Biochemistry & Molecular Biology, University of Massachusetts, Amherst, MA 01003, USA; vierling@biochem.umass.edu

**Keywords:** thioredoxins (TRXs), reactive nitrogen species, *S*-nitrosation, posttranslational modifications, *Arabidopsis thaliana*, *S*-Nitrosoglutathione reductase (GSNOR), glutaredoxins (GRXs), aldo-keto reductases (AKRs)

## Abstract

Protein cysteines (Cys) undergo a multitude of different reactive oxygen species (ROS), reactive sulfur species (RSS), and/or reactive nitrogen species (RNS)-derived modifications. *S*-nitrosation (also referred to as nitrosylation), the addition of a nitric oxide (NO) group to reactive Cys thiols, can alter protein stability and activity and can result in changes of protein subcellular localization. Although it is clear that this nitrosative posttranslational modification (PTM) regulates multiple signal transduction pathways in plants, the enzymatic systems that catalyze the reverse *S*-denitrosation reaction are poorly understood. This review provides an overview of the biochemistry and regulation of nitro-oxidative modifications of protein Cys residues with a focus on NO production and *S*-nitrosation. In addition, the importance and recent advances in defining enzymatic systems proposed to be involved in regulating *S*-denitrosation are addressed, specifically cytosolic thioredoxins (TRX) and the newly identified aldo-keto reductases (AKR).

## 1. Introduction

The molecular adaptation of plants in response to environmental cues is essential for maintaining proper metabolism and to ensure optimal plant fitness. One aspect of adaptation to biotic and abiotic stresses involves the posttranslational modifications (PTMs) of proteins. PTMs have emerged as important regulatory mechanisms in cell signaling and metabolism in all organisms, as they lead to alterations in protein activity, structure, and localization in order to respond to new cellular demands [1,2,3,4,5]. Oxidation and reduction (redox) reactions of specific amino acid side chains in proteins are a result of exposure to nitro-oxidative stresses including extreme temperatures, high light, drought, salinity and pathogen infection [6,7,8]. While the oxidation of amino acids such as His, Lys, Tyr and Trp are considered irreversible, reactive oxygen (ROS) and reactive nitrogen species (RNS) derived PTM of the sulfur-containing amino acids methionine (Met) and cysteine (Cys) are in most cases reversible [9,10]. In the past decades, more evidence has accumulated indicating that nitro-oxidative PTMs of cysteines are involved in a wide range of plant processes, including germination [5,11,12], root growth [13] and fertility [14,15,16]. This review provides an overview of nitro-oxidative protein modifications of Cys residues with a focus on protein *S*-nitrosation and summarizes the importance of enzymatic systems involved in the *S*-denitrosation reaction.

## 2. Protein Cys Thiols and Their ROS, RSS and RNS Dependent Modifications

Among the 20 common amino acids in proteins, Cys is one of the least abundant, but it is frequently observed in functionally important sites of proteins [17]. Cys in its protonated state is a neutral, polar amino acid that is involved in a diversity of functions, including structural stabilization, metal coordination and the regulation of protein activity. Structural Cys residues form intra- or intermolecular disulfide bonds in proteins often found in endomembrane system-associated or extracellular proteins [18]. Metal binding Cys residues, where ions such as iron, copper, nickel, zinc, or manganese are coordinated, can stabilize the structure or the bound ion can be involved in direct catalysis as a cofactor [19,20]. Redox-active regulatory thiols can be found as conserved residues in protein catalytic sites or in regions distinct from the active site and participate in several mechanisms like thiol-disulfide exchange reactions, for example in thioredoxins (TRXs), glutaredoxins (GRXs) and peroxiredoxins (PRXs), or in electron transfer reactions in the enzymes glutathione reductase (GR) and thioredoxin reductase (NTR) [21,22].

Among these classes of Cys residues, redox dependent Cys have been investigated for decades due to their ability to regulate protein function as a result of nitro-oxidative PTMs. The acid dissociation constant (pK_a_) of a particular protein Cys thiol dictates its reactivity at physiological pH. Based on the structure and local charge environment, certain protein thiols have a lower pK_a_ and exist as thiolate anions (R-S^−^) at typical intracellular pH, resulting in higher reactivity towards electrophiles [23]. Examples for proteins with low pK_a_ values in their active site Cys are redox-active enzymes like GRXs and TRXs, with values ranging between 3.9 (AtGRXS12; [24]) and 6.6 (CrTRXh1; [25]). In contrast, glutathione (GSH), the major non-protein redox-buffer in cells, has reported pK_a_ values of 8.7 and 9.4, which would imply it has a rather low reactivity towards electrophiles at physiological pH, but this is counterbalanced by its high mM intracellular concentration and rapid enzymatic recycling by the GR-system [24,26].

The most well studied PTM of sulfur-containing amino acids is the oxidation of Cys by ROS [27]. Here, the activated thiol is oxidized to a sulfenic acid derivative (R-SOH) by ROS such as hydrogen peroxide (Table 1 and Figure 1a). R-SOH also reacts with other Cys containing proteins to form intra- or intermolecular disulfide bonds. In the absence of a second thiol, R-SOH is highly reactive, unstable, and acts as an intermediate for the formation of further, higher oxidation states. This hyperoxidation upon ROS exposure generates sulfinic (R-SO_2_H) and sulfonic (R-SO_3_H) acids. While oxidation states up to R-SO_2_H can be reduced enzymatically by TRXs and GRXs, no reduction system for overoxidized Cys (R-SO_3_H) has been identified [28,29,30]. Besides the TRX and GRX system, sulfiredoxins (SRX) have been identified as specific enzymes catalyzing the reduction of overoxidized (R-SO_2_H) peroxiredoxins [30,31]. These thiol modifications can have regulatory downstream effects, thereby altering cellular function in response to various stresses or developmental cues (reviewed in [32]).

Gaseous signaling molecules like hydrogen sulfide (H_2_S), have recently emerged as important regulators in animals and plants. *S-*Persulfidation (R-SSH) occurs through the reaction of an oxidized protein thiol (R-SOH) with reactive sulfur species (RSS, Table 1), which can have inhibitory or activating effects on proteins [36]. It is assumed that RSS such as H_2_S and its dissociation products hydrosulfide (HS^−^) and sulfide (S^2−^) are able to catalyze the persulfidation of Cys, but the active form of H_2_S *in planta* is not fully understood [37]. H_2_S is produced in plants mainly in chloroplasts via the sulfate-assimilation pathway and also in mitochondria during the synthesis of β-cyanoalanine as a result of the detoxification of cyanide [38]. A recent proteomic approach identified over 2000 proteins as targets for this PTM in leaves of *Arabidopsis*, linking persulfidated cysteines to important processes like carbon metabolism and stress responses, as well as growth and development [38]. RSS-dependent Cys modifications have also been associated with physiological processes such as seed germination [39], fruit ripening, (reviewed in [40]) and stomatal movement [41]. Interestingly, proteins that undergo *S*-persulfidation are also prone to be regulated by modifications such as *S*-nitrosation, suggesting there is crosstalk between nitro-oxidative PTMs that may fine tune plant metabolism. Peroxisomal catalase (CAT), for example, an important enzyme in redox homeostasis, is inhibited by *S*-persulfidation as well as *S*-nitrosation [42,43], whereas ascorbate peroxidase is inhibited upon RSS-dependent modifications, but its activity is upregulated after *S*-nitrosation [44,45].

Alternatively, protein Cys can react with low-molecular weight thiols like GSH, resulting in mixed-disulfides, a reaction termed *S*-glutathionylation. It has been suggested that *S*-glutathionylation is a mechanism to protect critical Cys residues against overoxidation [46]. Some glutathionylated proteins exhibit increased activity, while others show a decrease. For example, stress induced *S*-glutathionylation of TRXf, a key component in redox regulation of chloroplastic carbon fixation, impairs light activation of target enzymes, slowing metabolism [47]. *S*-glutathionylation can occur either by the reaction of R-SOH with reduced GSH or with activated thiols (R-S^−^) and oxidized glutathione (GSSG). Another route is the reaction of thiols with *S*-nitrosoglutathione (GSNO), the product of RNS with GSH, providing further evidence for a direct crosstalk between ROS- and RNS-dependent signaling pathways. Deglutathionylation reactions are catalyzed enzymatically by GRX and SRX [24,48], but there is also evidence that atypical TRX proteins from poplar and yeast (TRX-like, TRX-lilium) are involved in the regulation of this PTM [49,50].

Nitric oxide (^•^NO; further referred to as NO) is a highly reactive gaseous molecule belonging to the RNS family due to unpaired electrons in its orbital (Table 1). It is a free radical that can gain or lose an electron, which leads to the formation of either the nitrosium cation (NO^+^) or the nitroxyl radical (NO^−^) [51]. NO is also a lipophilic molecule that diffuses through membranes [52] and has a half-lifetime of a few seconds. Therefore, NO rapidly reacts with other molecules, such as O_2_^•−^, metallo enzymes and O_2_. The reaction with O_2_ leads to the formation of nitrogen dioxide (NO_2_), which decomposes to nitrite and nitrate in aqueous solutions [53]. The major source of NO in animals is through an oxidative pathway catalyzed by nitric oxide synthases (NOS). Three isoforms, iNOS (inducible NOS), eNOS (endothelial NOS) and nNOS (neuronal NOS), catalyze the NADPH-dependent oxidation of L-arginine to N-hydroxy-arginine, following citrulline and NO formation by an oxidation step [53]. In land plants, however, no NOS enzyme has been identified, although arginine-dependent NOS-like activities in plants are sensitive to mammalian NOS inhibitors [52,54]. More recently, a cyanobacterial NOS enzyme was identified with high similarity to animal NOS, but with a different domain architecture [55]. This finding may lead to the identification of NOS-like enzymes in other photosynthetic organisms.

In higher plants, NO is produced in different compartments (cytosol, chloroplast, mitochondria, apoplast) through enzymatic and non-enzymatic mechanisms in reductive pathways. Non-enzymatically, it can be generated by carotenoids in the presence of light through the conversion of nitrogen dioxide [56] or induced by abscisic acid and gibberellins in the apoplast under acidic conditions with nitrite as a source [57]. The origin of enzymatic derived NO in plants, however, remains controversial. The best described enzymatic source of NO in plants is nitrate reductase (NR). *Arabidopsis* has two NR genes, NIA1 and NIA2, that are involved in nitrogen assimilation by reducing nitrate to nitrite. They also have been shown to catalyze nitrite-dependent NO formation in vitro and in vivo involving their molybdenum cofactor (Moco)-containing site in the *N*-terminal domain [58], similar to other members of the Moco-enzyme family such as sulfite oxidase (peroxisome), xanthine oxidoreductase/dehydrogenase (peroxisome), aldehyde oxidase (cytosol) and the amidoxime-reducing component (mitochondria) [59]. Recently, Santolini et al. [54] discussed the importance and contribution of NRs in NO production in plants. They concluded that NR-induced NO synthesis is limited by the availability of nitrite, which, under non-stressed conditions, is poorly concentrated in plant tissues due to the activity of plastidic nitrite reductase (NiR). Therefore, only under conditions where NiR is inhibited would NRs be able to generate NO from nitrite. Another group working on *Chlamydomonas* proposed an additional model where NR together with NADPH and an amidoxime-reducing component (later renamed to NO-forming nitrite reductase (NOFNiR) in higher plants) might represent a major system for NO synthesis in photosynthetic organisms [60]. In that model, NR would gain its electrons from NADPH and further provide those to NOFNiR, which displays a higher affinity for nitrite than NR. Current data highlight that the nitrogen assimilation pathway with its key metabolites nitrate and nitrite constitute major substrates for NO production in plants through enzymatic and non-enzymatic processes. However, further investigations are needed to determine the precise impact of nitrogen metabolism on NO homeostasis with respect to different physiological and developmental questions.

NO affects many plant processes, including plant defense [61,62], stomatal movement [63,64], flowering and fertility [14,15,65], plant-microbe interactions [66], and germination [67]. In addition, certain biotic and abiotic stresses induce NO production, linking NO to plant hormone homeostasis including salicylic and jasmonic acid signaling [68,69] and ethylene as well as auxin metabolism [70,71]. The major signaling and regulatory effect of NO and other RNS is through the reversible *S*-nitrosation (R-SNO) of critical Cys residues analogous to ROS and RSS-mediated modifications (Figure 1b). In addition, Tyr nitration and the nitrosylation of metal-containing proteins are also common PTMs associated with nitro-oxidative stresses (see [72] on terminology for NO-mediated PTMs). However, NO exhibits poor oxidant capacity under physiological conditions, therefore, nitrosative PTMs rely on molecules that are derived from the oxidation of NO with oxygen or ROS.

*S*-nitrosation events can be divided in general into four mechanisms, an oxidative pathway, a radical pathway, metal-transnitrosation and Cys-Cys transnitrosation [73,74]. In the oxidative pathway, RSNO is formed upon the reaction of thiolates with (auto)oxidation products of NO, such as nitrosonium cations (NO^+^), dinitrogen trioxide (N_2_O_3_) and nitrous acid (HNO_2_). The radical pathway comprises the reaction of peroxinitrite (ONOO^−^) with protein Cys residues, or through the reaction of NO radicals with Cys thiyl radicals (RS^−^). Metal-transnitrosation is a transition metal catalyzed pathway where NO binds, for example, to a protein ferric heme group, and the resulting Fe^3+^-NO complex then reacts with a thiolate to form R-SNO and ferrous heme [75]. NO can be further transferred from R-SNO through Cys-Cys transnitrosation reactions in which a thiolate attacks the nitrogen atom of the nitrosothiol, resulting in a nitroxyl disulfide intermediate that further decays to an RS^−^ and a newly formed R-SNO [75].

NO also reacts with GSH to form *S*-nitrosoglutathione (GSNO), which functions as a mobile NO reservoir in planta and is involved in the R-SNO formation of proteins in vitro and in vivo [20,76]. *S*-nitrosation has received increasing attention as an important nitro-oxidative regulatory mechanism in biological systems. The significance of this PTM in animal systems is well established, while in comparison, knowledge in plants is limited. Methodological developments of in vivo and in vitro labeling strategies, however, together with sensitive detection by mass spectrometry, will further allow for the identification of low abundant proteins that are modified by nitro-oxidative PTMs. Applying advanced approaches in diverse cells and tissues subjected to different conditions will facilitate linking these PTMs to developmental and environmental processes in plants.

## 3. Enzyme Catalyzed Regulation of *S*-Nitrosated Proteins

*S*-nitrosation is a reversible PTM of protein Cys residues that is involved in multiple plant processes as discussed above. While there has been considerable recent progress on the formation and identification of R-SNOs, less is known regarding the denitrosation reaction of this important modification. More and more evidence has accumulated in both animals and plants that *S*-nitrosation is reversed by specific proteins through either direct or indirect reaction mechanisms. In addition, GSH has been reported to effect the non-enzymatic *S*-denitrosation reaction of target proteins (reviewed in [77]), such as *Arabidopsis* GSNOR and GAPDH [78,79]. However, in the following, we discuss the importance and summarize recent findings on cytosolic thioredoxins, which are a specific subgroup of proteins that directly catalyze the denitrosation reaction and focus on enzymes that regulate the *S*-nitrosation status of proteins in an indirect manner.

### 3.1. Direct Enzyme-Catalyzed Denitrosation Reactions: Focus on Thioredoxins

#### 3.1.1. Thioredoxins in Photosynthetic Organisms

TRXs were first described by Peter Reichard and coworkers in the 1960s as small, heat stable protein cofactors required for activity of the essential enzyme ribonucleotide reductase [80]. TRXs are ubiquitous, multifunctional thiol-disulfide oxidoreductases containing two Cys residues in a conserved active site motif (WC[G/P]PC) [49]. TRXs are components of an important thiol antioxidant system consisting of TRX and NADPH-dependent thioredoxin reductases (NTR) which act to regenerate reduced TRX Cys residues. Together with other TRX-superfamily members, such as GRXs, protein-disulfide isomerases, glutathione peroxidases and glutathione-*S*-transferases, they share a common structural motif, the thioredoxin fold, that consists of five stranded β-sheets and four flanking α-helices [81,82]. The nucleophilic Cys residue of TRX, located at the *N*-terminal side of the active motif, is deprotonated even under physiological conditions and largely exposed, allowing for direct protein-protein interaction through intermolecular disulfide bonding [21]. The second resolving Cys is mostly buried and usually protonated. It functions by attacking the intermolecular disulfide bond, releasing the reduced target protein. In plants, the regeneration of TRXs is catalyzed by NADPH-dependent NTRA or B in the cytosol and mitochondria, respectively, and by NTRC and ferredoxin-thioredoxin reductase in chloroplasts [83,84].

The functions of different members of the TRX superfamily largely depend on varying structural features and expression patterns, but also on their specific redox midpoint potential (Em) [85]. The Em is a characteristic feature particularly determined by the two amino acids between the two Cys residues and the surrounding microenvironment [25]. Reported redox potentials within the TRX superfamily vary from −280 mV for TRXh2 from *Arabidopsis* to −120 mV for DSBA from *E*. *coli* [86,87]. At present, plants possess the largest family of TRXs with 20 members in *Arabidopsis* that can be divided into subgroups, those found in the chloroplast: TRX-m1, 2 3 and 4, TRX-f1 and 2, TRX-x, TRX-y1 and 2 and TRX-z; the mitochondria: TRX-o1 and 2; and the eight member TRX-h group, primarily found in the cytosol (Table 2) [83,88]. In addition, more than 40 TRX-like proteins have been identified [49,89]. The evolutionary origin of the TRX m, x and y types is prokaryotic, whereas types f, h and o are of eukaryotic origin [88]. The diversity of isoforms seems to provide plants with an additional antioxidant system compared to mammals where only two types of TRX have been described: TRX1 in the cytosol and the mitochondrial TRX2 [90].

#### 3.1.2. The Cytosolic H-Type TRXs

The cytosolic TRX-system consists of h-type TRXs and has been studied in *Arabidopsis* and other plants, including rice, *Medicago* and poplar [83,92,97]. The *Arabidopsis* genome encodes nine TRXh genes that are distributed across chromosomes 1, 3 and 5 (Figure 2a). However, one of the genes, *TRXh10* (AT3G56420), is listed as putative in the Uniprot database and therefore is not included in this review. Further analysis is required to show whether the *TRXh10* gene is actively expressed or not. Genomic sequences of the h-type TRXs range between 634 (*TRXh7)* and 1225 bps (*TRXh8),* with a similar gene-structure split into three or four exons, indicating that all h-type TRXs may have evolved from a common ancestor gene (Figure 2b).

All h-type TRXs have a WC[G/P]PC active site in a single TRX domain (Table 2 and Figure 3) and can be classified into three subgroups based on their phylogenic relationship [83]. Members of the subgroup I are located in the cytosol and exhibit insulin reduction activity with NTR and DTT as electron donors [91,93,95]. TRXh2, in subgroup II, also shows insulin reduction activity, but is reported to be localized in both the cytosol and mitochondria [85]. The other type II and III proteins are less well characterized. However, the *Medicago truncatula* TRXh7 and TRXh8 orthologs also show insulin reduction, but to a lesser extent [97]. Cys exchange variants of the *N*-terminally located Cys residues of TRXh3, which also is present in all other members of the subgroup I and additionally in TRXh8 (Figure 3), showed that this cysteine residue is not involved in the catalytic activity [99]. Interestingly, some isoforms (h2, h7, h8 and h9) are associated with the endomembrane system, most likely due to myristoylation at their *N*-terminal extensions (glycine residue at position 2) [100,101].

Some studies have provided evidence that h-type TRXs are involved in processes such as mitochondrial metabolism [102], calcium signaling [103], and germination [104,105], but only a few functions have been associated with specific TRXh isoforms based on direct protein-protein interactions or genetic studies. Different TRXhs are able to reduce homodimerized cytosolic malate dehydrogenase in vitro and in planta, thereby maintaining redox-homeostasis by minimizing the oxidative inactivation of MDH [106]. TRXh1 is involved in cyanide detoxification in *Arabidopsis* upon interaction with sulfurtransferases [107] as well as in the fine-tuning of phosphate metabolism through regulation of the E2 ubiquitin conjugase PHO2 together with TRXh4 [108]. In addition, TRXh1 is involved in modulating ROS production in *Arabidopsis* under anoxic conditions by interacting with HRU1 (hypoxia responsive universal stress protein 1) [109].

Other plant metabolic processes are regulated by the TRXh2 isoform. Fonseca-Pereira and coworkers [110] reported that TRXh2 contributes to the redox regulation of mitochondrial photorespiratory metabolism through direct deactivation of glycine decarboxylase (GDC-L) in vitro. TRXh3, on the other hand shows a redox-dependent functional switch from a disulfide reductase to a molecular chaperone under heat stress [111]. In addition, TRXh3 is, in conjunction with TRXh5, involved in the monomerization of NPR1 in response to pathogens and cold acclimation [96,112]. TRXh3 is also involved in glutathione metabolism by reducing GSSG in vitro [113], a function that has previously been assigned to the GRX system rather than TRXs. However, the overall reduction rate of GSSG by TRXh3 together with NTRA was ~200-fold lower in comparison to the GRX system using GR1, therefore, further experiments are needed to show the significance of TRXh3 in this process, especially in vivo.

There are no publications on the specific function of TRXh4. Nevertheless, TRXh4 is highly expressed in developing embryos and dry seeds based on the ePlant database, indicating that it might be involved in fertility and germination. Besides its oxidoreductase activity, TRXh5 was identified as a key regulator in the systemic acquired resistance signaling by specific *S*-denitrosation of NPR1 [96,114], and plants lacking TRXh5 show a higher susceptibility to pathogens. In addition, TRXh5 is associated in the plant tolerance to the fungal pathogen *Cochliobolus victoriae* [115], further indicating a specific role for this isoform in disease- and pathogen-related processes.

TRXh7 is, as mentioned earlier, associated with the endomembrane system due to its *N*-myristoylation at the *N*-terminus. Recent work from Baune et al. [116] could show that TRXh7 interacts with the glucose-6-phosphate (G6P)/phosphate translocator GPT1 and is involved in the redox-dependent release of the transporter to peroxisomes, thereby regulating the oxidative pentose phosphate pathway. Interestingly, GPT1 regulation presumably involves *S*-glutathionylation of Cys65 in the GPT1 N-terminus, which in turn could be regulated by cytosolic GrxC1, which is also reported to be posstranslationally associated to endomembranes upon *N*-myristoylation.

There have been no studies on the specific function of TRXh8 and its involvement in nitro-oxidative or stress related processes. TRXh9 shows, in contrast, a remarkable redox-dependent characteristic. Two amino acids in its *N*-terminal extension (Gly2 and Cys4), are important for the association with plasma membranes, due to a combination of Gly myristoylation and Cys palmitoylation [98]. Both modifications together are important for TRXh9 to move from cell to cell, implicating an involvement in intercellular communication. In addition, Cys4 and Cys57 of TRXh9 have been identified as important residues in the dithiol-disulfide exchange reaction and the subsequent reduction of oxidized glutathione peroxidase 3 (GPX3) [117].

#### 3.1.3. TRXs Can Denitrosate Proteins via Two Proposed Mechanisms

In general, TRXs are well characterized oxidoreductases that catalyze the reduction of oxidized target proteins [88,89]. More recently, however, they have been reported to also be involved in the specific denitrosation reaction of protein-SNOs. Besides human TRX1 and TRXh1 from *Chlamydomonas*, *Arabidopsis* TRXh5 has been described as an important enzyme controlling the *S*-nitrosation status of proteins [96,118,119]. TRXs are proposed to catalyze the *S*-denitrosation reaction of target proteins via two distinct mechanisms [96]. In a reductive pathway, the nucleophilic Cys residue of TRX displaces NO from the target Cys by heterolytic cleavage, forming an intermolecular disulfide bond between TRX and its target substrate. Subsequently, the resolving active site Cys attacks the mixed disulfide and gets oxidized, releasing the reduced substrate. Oxidized TRX is then recycled by NTR using NADPH as the electron donor (Figure 4a). The second mechanism by which nitro-oxidative PTMs can be regulated is demonstrated by emerging data demonstrating Cys-Cys transnitrosation reactions—the transfer of an NO group from one R-SNO to the free thiol of another protein. (Figure 4b) [120]. For example, HsTRX1 is able to catalyze transnitrosation from its non-active site Cys73 to Caspase 3, leading to an inhibition of caspase activity [121]. A Cys-Cys transnitrosation reaction has also been discussed as one potential mechanism by which TRXh5 regulates the RNS-derived PTM of target proteins [96].

No data on denitrosation activities for other h-type TRXs are published. Given the high amino acid sequence identity of TRXh5 with other cytosolic TRX isoforms from *Arabidopsis* (≥60%; Table 3), it could be speculated that the other class I members also denitrosate target proteins. In comparison to the known human denitrosation enzyme, TRX1, the percent sequence identity/similarity is even lower, indicating that *Arabidopsis* class II and III h-type TRX might also show denitrosation activities. Kneeshaw and colleagues [96] reported that TRXh5 shows selective denitrosation activity of specific target proteins. Assuming a spatiotemporal expression of the cytosolic TRXs, it could be that specific TRX isoforms target different substrates. For example, membrane bound TRXh2, h8 and h9 could denitrosate other membrane-bound proteins, which would allow further NO-related signal propagation. Furthermore, differences in TRX sequences may also dictate interaction with different substrates. Additional experiments are necessary to assess the involvement of other cytosolic TRXs in denitrosation reactions, their substrate specificity and their role in general NO homeostasis in *Arabidopsis* and other plants.

### 3.2. Other Direct Enzymatic Denitrosation Systems

Other oxidoreductases have been described as direct denitrosation enzymes in mammals. For example, human dithiol GRX1 and monothiol GRX5 exhibit denitrosation activity in vitro with nitrosated caspase 3 and cathepsin B as substrates [122]. The mechanism is similar to the proposed TRX mechanisms, where either one or both cysteines of the oxidoreductase are involved (Figure 4). However, in contrast to the TRX system, GRXs are dependent on GSH and glutathione reductase for their activity [28]. Sulfiredoxins, oxidoreductases initially reported as enzymes involved in the reduction of hyperoxidized proteins, have recently been identified as specific enzymes that catalyze the ATP-dependent reduction of nitrosated peroxiredoxin 2 (PRX2) in humans [123,124]. No GRX- or SRX-dependent denitrosation systems have been reported in plants, but given the general conservation of those enzymes among species, some plant GRXs or SRXs might also be involved in denitrosation reactions.

### 3.3. Indirect Enzyme-Catalyzed Denitrosation Systems

Besides the direct denitrosation of target proteins, there are enzymatic systems that regulate nitrosation status indirectly. Reduction of GSNO by the enzyme *S*-nitrosoglutathione reductase (GSNOR) is a major route of GSNO catabolism in all organisms [15,125]. This highly conserved enzyme is present as a single copy gene in most higher plants and is expressed ubiquitously in the cytosol and nucleus [14,15]. The mutation of *Arabidopsis* GSNOR (AT5G43940) leads to higher intracellular concentrations of protein-SNOs, demonstrating the critical role of this enzyme in plant NO homeostasis. In addition, T-DNA null insertion alleles show multiple plant growth defects, including shorter and multibranching inflorescences, reduced lateral roots, compromised pathogen response and reduced fertility [15,126,127,128], demonstrating that a fine-tuned NO homeostasis is mandatory for proper plant development. GSNOR acts as a homodimer that catalyzes the NADH-dependent reduction of GSNO to *N*-hydroxysulfinamide (GSNHOH), which through spontaneous downstream reactions results in the production of GSSG and NH4^+^ (reviewed in [129]). Interestingly, GSNO and other nitroso compounds have been identified as inducing S-nitrosation of *Arabidopsis* GSNOR at specific solvent accessible Cys residues, negatively affecting GSNOR activity, which may allow for proper NO-dependent signal transduction [20,130,131]. More recently it was reported that catalase 3 is able to trans-nitrosate GSNOR at Cys10, leading to structural alterations and GSNOR targeted degradation through autophagy [131].

Although GSNOR has been acknowledged as a critical enzyme controlling NO homeostasis and thereby regulating the *S*-nitrosation status of proteins in plants and other organisms, recent data indicate that a specific mammalian aldo-keto reductase AKR1A1 also plays a role in controlling GSNO levels [132]. AKR1A1 is upregulated in GSNOR-deficient mice, presumably as a compensatory mechanism [132,133]. AKRs comprise a superfamily of generally monomeric 34–37 kDa oxidoreductases that share a common (α/β)8-barrel structural motif and act to decompose a broad range of reactive carbonyl substrates produced during stress [134,135,136]. The substrate specificity is determined by three structural loops and a conserved catalytic tetrad consisting of Asp, Tyr, Lys and His, which is essential for the enzymatic activity of these proteins [137]. In contrast to GSNOR, which is NADH-dependent, the GSNO and S-nitrosated coenzyme A (SNO-CoA) reduction activity of AKR1A1 is dependent on NADPH as cofactor [135,138]. AKRs in general catalyze an ordered bi kinetic mechanism in which NADPH binds first and leaves last [139]. GSNO reduction by AKR1A1 follows the canonical AKR reaction scheme, where the hydride transfer from NADPH to the nitrogen atom of the SNO moiety and protonation of the oxygen atom by the active site Tyr generates a *S*-(*N*-hydroxy) intermediate that rearranges eventually to GSH sulfinamide [132].

Plants encode a large number of aldo keto reductases with *Arabidopsis* having 22 AKR proteins (Figure 5). Although no specific plant orthologue of the human AKR1A1 protein can be identified, *Arabidopsis* AKR4C8 is the homolog with the most similar amino acid sequence to human AKR1A1. AKR4C8 is in a clade with three other cytosolic AKRs in a 4C subclass (AKR4C9, AKR4C10, AKR4C11), with 64–82% sequence identity and 73–86% similarity, respectively (Figure 5).

Recent research identified that AKR4C8 and AKR4C9 are upregulated in *Arabidopsis* GSNOR-null mutant leaves [140]. Furthermore, it could be demonstrated that all four members of this subclass catalyze the NADPH-dependent reduction of GSNO, although less efficiently than GSNOR [140]. In addition, plants lacking GSNOR show increased NADPH-dependent GSNO reduction in planta [140]. These data support the hypothesis that these AKR4C proteins are additional components regulating the NO homeostasis in plants. However, it should be noted the AKR4C proteins have also been characterized as having significant activity with other substrates [137,141]. Defining the most significant in vivo substrates of these diverse and typically promiscuous enzymes requires further analysis, and whether other AKRs participate in NO homeostasis through reduction of GSNO is an open question. The catalytic tetrad is mostly conserved among all *Arabidopsis* AKRs (Figure 6), with the exception of the ALKR group, where the positively charged Lys residue is exchanged with a negatively charged Glu. However, there is high structural variation in the three loops that define substrate binding, indicating that there are differences in substrate recognition between the AKR proteins. Further work will be necessary to integrate the newly identified AKR proteins in the network controlling NO homeostasis in plants and other organisms (Figure 7).

## 4. Concluding Remarks and Future Outlook

Despite the involvement of NO in multiple plant processes, including germination, root growth and fertility, a basic understanding of the mechanisms by which NO exerts its effects is lacking. NO and its derivatives can affect physiological processes through the reversible *S*-nitrosation of critical protein cysteines, which regulates protein activity, structural stability and localization. Past research on TRXs has mainly focused on biochemical characterization and their specificities for different target enzymes with respect to ROS related PTMs, but recent studies identified that a specific subset of cytosolic TRXs in different organisms are able to catalyze the direct denitrosation of *S*-nitrosated target proteins. Future studies will be necessary to address the denitrosation activity of other cytosolic TRX isoforms using purified proteins as well as by generation of plant mutants for single and multiple TRXs. There are no studies on chloroplast or mitochondrial TRX enzymes in plants with regard to *S*-denitrosation reactions. However, given that chloroplasts as well as mitochondrial proteins are known targets for nitro-oxidative modifications, it remains to be elucidated whether organellar TRX systems are also involved in catalyzing the *S*-denitrosation of target proteins. Evidence in mammalian systems further suggests that GRXs as well as sulfiredoxins may form an additional, cooperative system regulating NO-derived PTMs. GSNOR is considered a key regulator of NO homeostasis through NADH-dependent catabolism of GSNO, the bioactive form of NO, fine-tuning the *S*-nitrosation status of proteins indirectly. AKRs represent a newly identified, additional system capable of regulating GSNO in an NADPH-dependent manner. Biochemical studies to further elucidate the AKR catalytic mechanism and substrate affinities as well as genetic analyses with plant mutants are needed to show their significance in regulating GSNO levels and overall NO homeostasis.

## Figures and Tables

**Figure 1 antioxidants-11-01411-f001:**
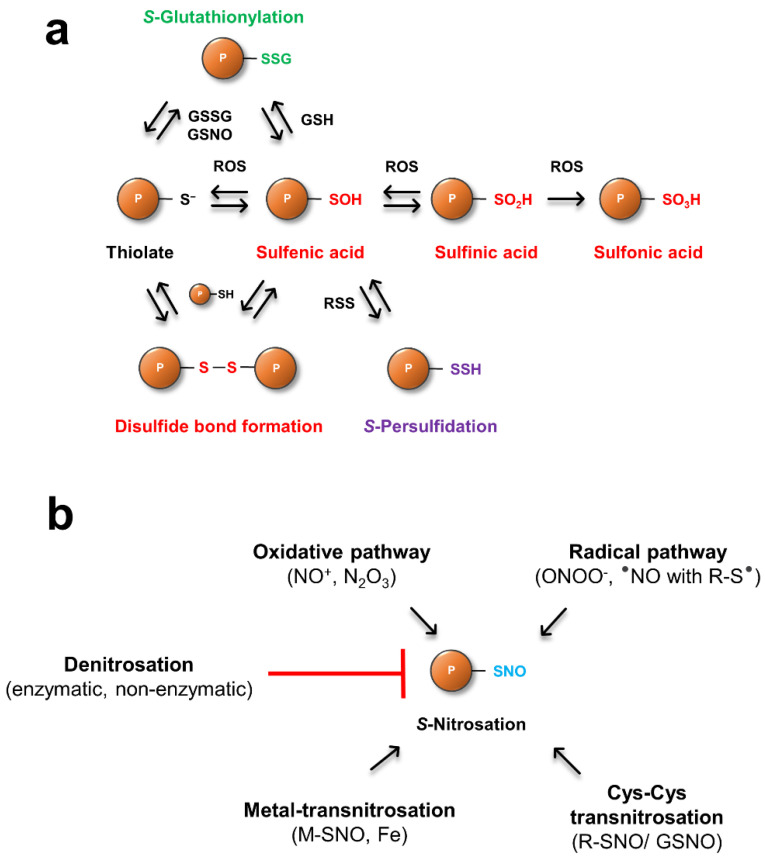
Mechanisms of ROS, RSS (**a**) and RNS (**b**) dependent posttranslational modifications of protein Cys residues. See text for details.

**Figure 2 antioxidants-11-01411-f002:**
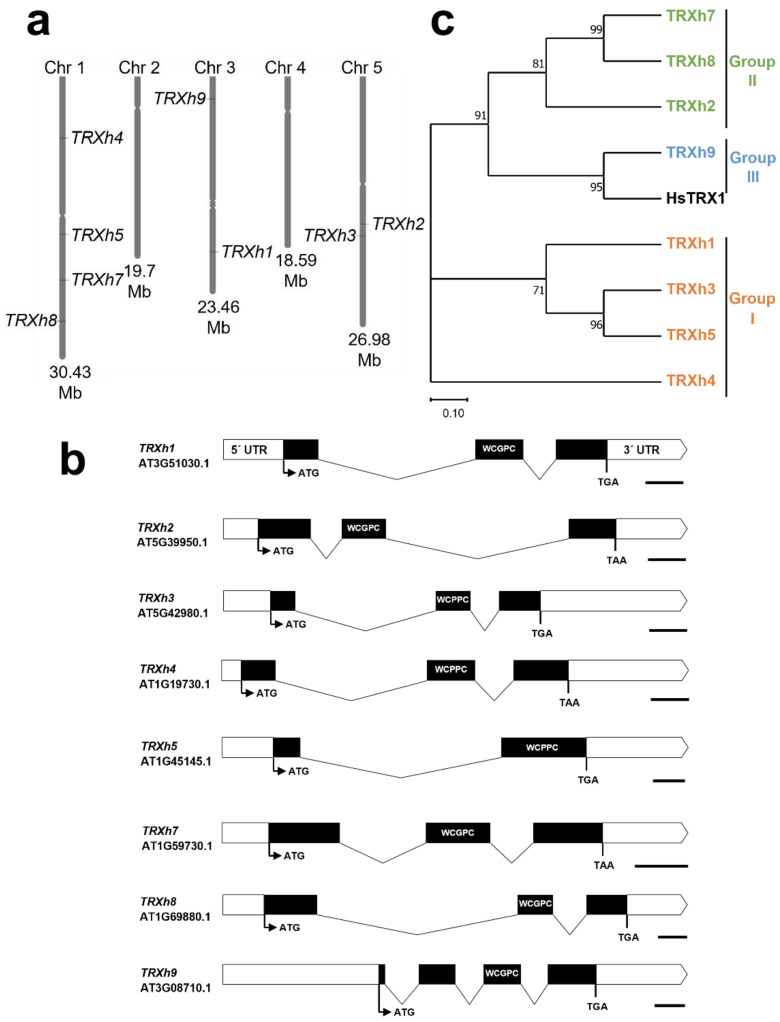
Chromosomal location, gene structure and phylogenetic relationships of h-type thioredoxins from *Arabidopsis*. (**a**) Chromosomal location of h-type TRXs from *Arabidopsis*. (**b**) Intron-exon structure of representative gene models. Exons are represented by black bars and introns by folded black lines, while 5′- and 3′ UTR regions are white boxes. Lines and bars to scale and represent total sequence length. ATG; start codon, TGA/TAA; stop codon, WCG/PPC; active site. Scale bars represent 100 nucleotides. (**c**) Maximum likelihood phylogenies for h-type TRX proteins from *Arabidopsis* and TRX1 from *Homo sapiens*. Protein sequences from representative gene models were used with the MEGA 11 program with bootstrap test (1000 times) and neighbor-joining method. Branch lengths are proportional to phylogenetic distances.

**Figure 3 antioxidants-11-01411-f003:**
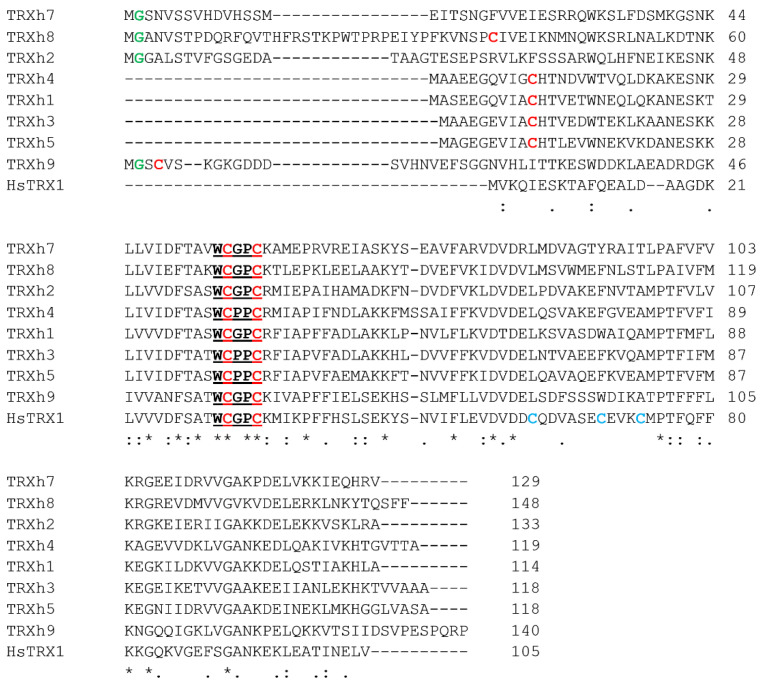
Conserved features of *Arabidopsis* h-type TRXs and human TRX1 protein sequences. Highlighted in red are the active site Cys residues as well as the *N*-terminal Cys. The catalytic motif is underlined and additional *C*-terminal Cys residues unique to HsTRX1 are in blue. Glycine residues that undergo *N*-myristoylation are in green. Asterisks (*) denote conserved residues, (:) strong and (.) weakly similar amino acid properties.

**Figure 4 antioxidants-11-01411-f004:**
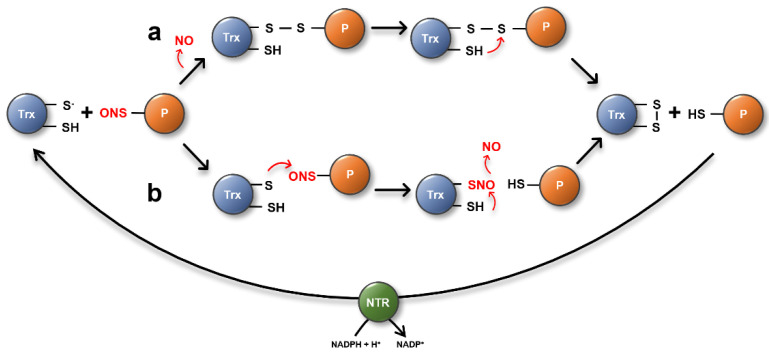
Proposed TRX denitrosation mechanisms via the reductive (**a**) or transnitrosation pathway (**b**). In the reductive pathway (**a**), the nucleophilic Cys residue of the TRX displaces NO from the target Cys by heterolytic cleavage, resulting in the formation of an intermolecular disulfide bond between TRX and its target substrate. Subsequently, the resolving active site Cys attacks the mixed disulfide and gets oxidized, releasing the reduced substrate. Oxidized TRX is then recycled by NTR. (**b**) R-SNOs can undergo a transnitrosation reaction with another thiol leading to the transfer of NO.

**Figure 5 antioxidants-11-01411-f005:**
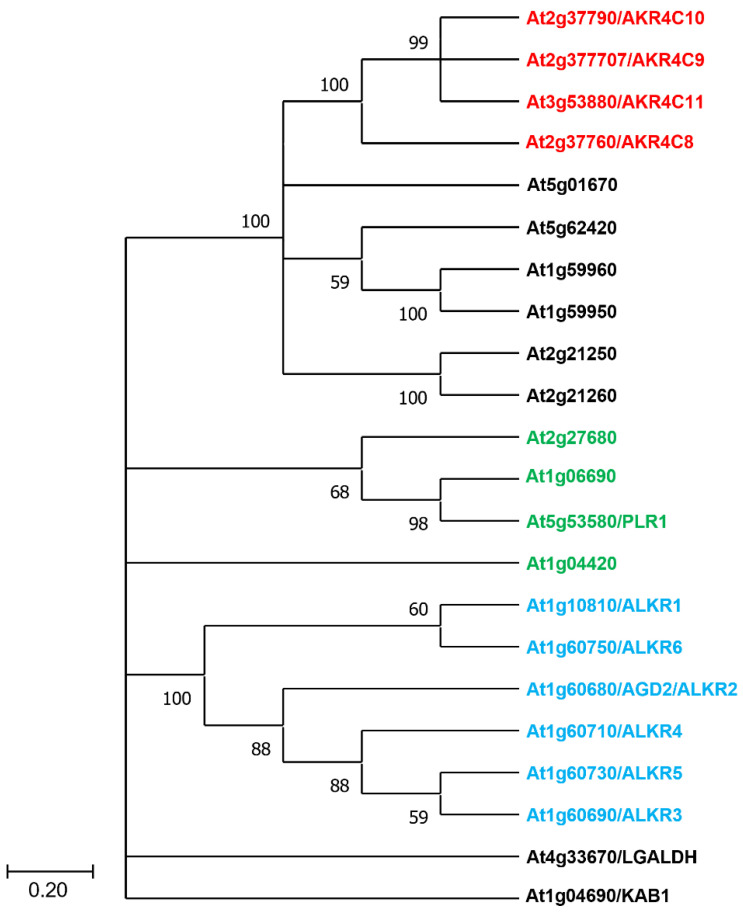
Phylogenetic tree of Arabidopsis AKR proteins. The phylogenetic tree was constructed by searching for aldo-keto reductases (PTHR11732) in the PANTHER database and using the MEGA 11 program with bootstrap test (1000 times) and the neighbor-joining method. Highlighted in red are the four members of the subclass 4C, while marked in blue are AKR proteins that lack the Lys residue in their catalytical tetrad. Green denotes AKR proteins that show an *N*-terminal extension, indicating they may localize to chloroplasts or mitochondria. In addition to the AGI identifiers, other names given to each AKR are shown.

**Figure 6 antioxidants-11-01411-f006:**
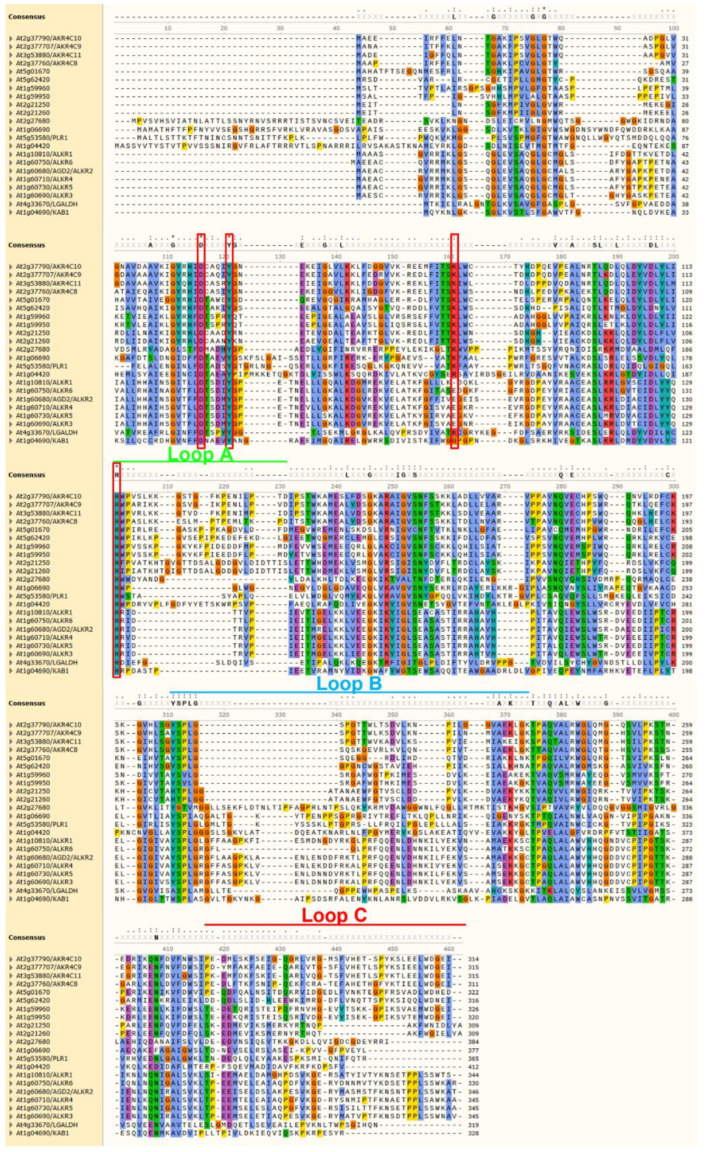
Multiple sequence alignment of AKRs from *Arabidopsis*. Red boxes highlight the catalytic tetrad residues, while green, cyan and red bars above the alignment denote the flexible loops defining the active site important for substrate specificity. Elements were assigned using the structural information of *Arabidopsis* AKR4C8 (PDB code 3h7r). Residues are color-coded based on their properties: red: positive; blue: hydrophobic; green: polar; orange: glycine; purple: negative; teal: aromatic; yellow: proline.

**Figure 7 antioxidants-11-01411-f007:**
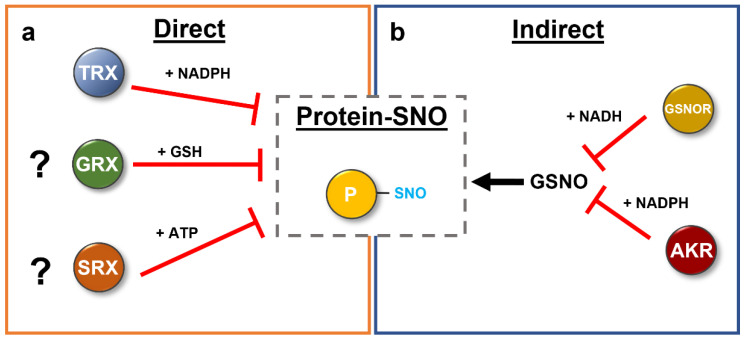
Direct (**a**) and indirect (**b**) denitrosation of proteins in plants. Enzymatic denitrosation systems have been reported in the literature. The thioredoxin system uses NADPH to remove the nitroso group from *S*-nitrosated target proteins via two distinct mechanisms. However, further studies have to show the contribution of the GRX and SRX-system in this process in plants. In contrast, the level of available NO is regulated by GSNOR and the newly identified AKR proteins, thereby modulating the *S*-nitrosation status of proteins indirectly. See text for further details.

**Table 1 antioxidants-11-01411-t001:** List of major reactive oxygen species (ROS), reactive nitrogen species (RNS), and reactive sulfur species (RSS). Adapted from [33,34,35].

ROS	RNS	RSS
^1^O_2_, singlet oxygen	^•^NO, nitric oxide	RS^•^, thiyl radical
H_2_O_2_, hydrogen peroxide	ONOO^−^, peroxynitrite	H_2_S, hydrogen sulfide (hydrosulfide (HS^−^) and sulfide (S^2−^))
O^•−^_2_, superoxide radical	NO_2_, nitrogen dioxide	RSOH, sulfonic acid and RSO_2_H, sulfinic acid
OH^•^, hydroxyl radical	N_2_O_3_, dinitrogen trioxide	RS(O)SR, disulfide-S-monoxide/thiosulfinate
HO_2_^•^, hydroperoxyl radical	N_2_O_4_, dinitrogen tetraoxide	RS(O)_2_SR, disulfide-S-dioxide/thiosulfonate

**Table 2 antioxidants-11-01411-t002:** *Arabidopsis* h-type Thioredoxins.

Protein	AGI	Uniprot ID	MW (in kDa)	Subgroup	Localization	Active Site Motif	Activity	References
TRXh1	AT3G51030	P29448	12.67	I	Cytosol	WCGPC	Insulin reduction (Ta-, Os- and AtTRXh1)	[91,92]
TRXh2	AT5G39950	Q38879	14.67	II	Cytosol, Mitochondria	WCGPC	Insulin reduction (TaTRXh2)	[91]
TRXh3	AT5G42980	Q42403	13.10	I	Cytosol	WCPPC	Insulin reduction (Ta- and AtTRXh3)	[91,93,94]
TRXh4	AT1G19730	Q39239	13.06	I	Cytosol	WCPPC	Insulin reduction (AtTRXh4); unpublished results	[95]
TRXh5	AT1G45145	Q39241	13.12	I	Cytosol	WCPPC	Insulin reduction (AtTRXh5) and S-denitrosation reaction	[95,96]
TRXh7	AT1G59730	Q9XIF4	14.53	II	Cytosol, Plasma membrane	WCGPC	Insulin reduction (MtTRXh7)	[97]
TRXh8	AT1G69880	Q9CAS1	17.25	II	Cytosol, Plasma membrane	WCGPC	Insulin reduction (MtTRXh8)	[97]
TRXh9	AT3G08710	Q9C9Y6	15.33	III	Cytosol, Plasma membrane	WCGPC	/	[98]

**Table 3 antioxidants-11-01411-t003:** Percent amino acid identity/similarity of *Arabidopsis* h-type TRXs and human TRX1. Highlighted in red are the percent identity/similarity of h-type TRX in comparison to TRXh5. Data were assessed using SIAS (http://imed.med.ucm.es/Tools/sias.html, accessed on 26 April 2022) and the EBLOSUM62 matrix with default parameters. Values were rounded to the nearest integer.

**TRXh1**	100								
**TRXh2**	43/59	100							
**TRXh3**	62/73	40/53	100						
**TRXh4**	60/71	45/58	61/72	100					
**TRXh5**	61/75	45/57	74/82	60/75	100				
**TRXh7**	38/49	42/55	36/50	40/54	39/49	100			
**TRXh8**	33/51	38/57	35/50	37/52	37/50	47/58	100		
**TRXh9**	44/56	31/47	37/48	39/51	36/53	35/45	26/38	100	
**HsTRX1**	43/56	41/54	41/52	46/55	36/55	36/47	33/47	46/59	100
	**TRXh1**	**TRXh2**	**TRXh3**	**TRXh4**	**TRXh5**	**TRXh7**	**TRXh8**	**TRXh9**	**HsTRX1**

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
