# Peer review of "Focus on Nitric Oxide Homeostasis: Direct and Indirect Enzymatic Regulation of Protein Denitrosation Reactions in Plants"

_antioxidants, 2022, doi:10.3390/antiox11071411_

Round 1

Reviewer 1 Report

The MS entitled “Focus on nitric oxide homeostasis: Direct and indirect enzymatic regulation of protein denitrosation reactions in plants” reviews S-nitrosation and how plants are balancing this type of PTM. The review is mainly focused on the role of cytosolic TRXs h, GSNOR and AKRs in plants, specifically in the model plant Arabidopsis thaliana.

In general, this work is well structured and written. Regarding the MS content, I have some comments:

1) It has been well established that TRXs have an active role in NO regulation, especially TRX h5. However, what is the role of the TRX system in other cell compartments? For instance, chloroplasts have an important redox regulation mediated by TRXs, and it is known that regulatory cysteines are prone to suffer different types of PTMs (oxidation, nitrosation and glutationylation). Is nitrosation more important in the plant cytosol, with respect to other sub-cellular compartments, or there are more TRX-h related studies because the first investigations came from animal research? Please, include this explanation in the MS.

2) In line 267, I think it is not correct to say that “…NTR and DTT as reducing systems”. Systems are composed of more than one element and, moreover, DTT is a non-physiological electron donor. Please, clarify this point.

3) If plants lacking GSNOR show increased NADPH-dependent GSNO reduction in planta (line 448), why is there a characteristic phenotype in gsnor mutants? If there is no a compensation effect, might these denitrosation enzymes (GSNOR and AKRs) have different additional functions? Please, develop or explain better the corresponding sentence.

Minor comments:

· A comma is needed after “S-denitrosation reaction” in line 16.

· Please, put in italics the letters referring to the type of TRX (and others, like in “S-denitrosation”).

· In line 443, “identity” instead of “identify”.

Author Response

We would like to thank both experts for their comments on our work. The manuscript in the current version was modified according to the suggestions of the reviewers, and we hope it increased the quality of our work.

Our responses are marked in red for easier evaluation.

The MS entitled “Focus on nitric oxide homeostasis: Direct and indirect enzymatic regulation of protein denitrosation reactions in plants” reviews S-nitrosation and how plants are balancing this type of PTM. The review is mainly focused on the role of cytosolic TRXs h, GSNOR and AKRs in plants, specifically in the model plant Arabidopsis thaliana.

In general, this work is well structured and written. Regarding the MS content, I have some comments:

  • It has been well established that TRXs have an active role in NO regulation, especially TRX h5. However, what is the role of the TRX system in other cell compartments? For instance, chloroplasts have an important redox regulation mediated by TRXs, and it is known that regulatory cysteines are prone to suffer different types of PTMs (oxidation, nitrosation and glutationylation). Is nitrosation more important in the plant cytosol, with respect to other sub-cellular compartments, or there are more TRX-hrelated studies because the first investigations came from animal research? Please, include this explanation in the MS.

Response: We would like to thank the reviewer for this suggestion. To our knowledge there are no publications that address the involvement of the organellar TRX system in S-denitrosation reactions. We therefore added the following to the manuscript (lines: 484-488):

There are no studies on chloroplast or mitochondrial TRX enzymes in plants with regards to S-denitrosation reactions. However, given that chloroplast as well as mitochondrial proteins are known targets for nitro-oxidative modifications, it remains to be elucidated whether organellar TRX systems are also involved in catalyzing S-denitrosation of target proteins.  

  • In line 267, I think it is not correct to say that “…NTR and DTT as reducing systems”. Systems are composed of more than one element and, moreover, DTT is a non-physiological electron donor. Please, clarify this point.

Response: “Systems” is not appropriate in this context and we therefore changed the sentence to: “Members of the subgroup I are located in the cytosol and exhibit insulin reduction activity with NTR and DTT as electron donors.

  • If plants lacking GSNOR show increased NADPH-dependent GSNO reduction in planta(line 448), why is there a characteristic phenotype in gsnor mutants? If there is no a compensation effect, might these denitrosation enzymes (GSNOR and AKRs) have different additional functions? Please, develop or explain better the corresponding sentence.

Response: We would like to thank the reviewer for this interesting comment, as we believe that investigating the potential compensation effect of the AKR4Cs is a topic worth considering in future work studies we are indeed working on it. Possible explanations are:

  1. AKR4Cs are less active than GSNOR, and since hot5-2 is considered a genotype accumulating RNS and SNOs it might be that AKR4Cs show nitro-oxidative sensitivity due to redox dependent modifications (S-nitrosation, S-glutathionylation, oxidation). Similar has been already observed for GSNOR, where different NO-donors (including GSNO) inhibit GSNOR activity.
  2. Substrate specificity and spatiotemporal expression of AKR4Cs: AKRs show a broad substrate specificity and we are currently analyzing what the preferred substrate of the different isoforms are. In addition, analyzing the precise expression pattern of the AKRs also with respect to different nitro-oxidative stresses will give a better understanding on additional functions of these proteins.
  3. Phenotype of AKR4C mutants. There are no studies on insertion alleles for the four AKR4C genes. Future work with single and higher order mutants will allow for comparison with similar phenotypes of genotypes that are known to be involved in NO homeostasis (e.g. hot5-2).

Minor comments:

  • A comma is needed after “S-denitrosation reaction” in line 16.

Response: We changed the sentence.

  • Please, put in italics the letters referring to the type of TRX (and others, like in “S-denitrosation”).

Response: We are following the general nomenclature recommended by TAIR (https://www.arabidopsis.org/portals/nomenclature/namerule.jsp), where protein products of genes should be written without italics. We changed S-nitrosation to S-nitrosation.

  • In line 443, “identity” instead of “identify”.

Response: Identify has been changed to identity.

Reviewer 2 Report

This review by Patrick Treffon and Elizabeth Vierling provides an interesting overview of the biochemistry of S-nitrosation and S-denitrosation of proteins in plants, with emphasis on thioredoxins h and aldo-keto reductases for their direct and indirect potential role in protein S-denitrosation. The review is well written and organized. I have only a short list of minor comments:

-       Line 53: I wouldn’t say that “Redox-active regulatory thiols are conserved in protein catalytic sites” without adding the information that redox-active regulatory thiols are quite often away from proteins active sites and yet responsible of the regulation of their activity.

-       Line 76: As far as I know, sulfiredoxins were shown to reduce sulfinic to sulfenic acids in peroxiredoxins only. It seems more an exception rather than a rule. The statement instead seems to suggest that TRXs, GRXs and SRX are equally effective.

-       Figure 1a might be improved. Some suggestions: GSSG and GSNO should be close to the arrow from P-S- to P-SSG (not the opposite one). H2O2 should be added above the arrow from P-S- to P-SOH. Maybe another arrow should be added to show that disulfide bonds may derive from thiolates, not only from sulfenic acids. In Figure 1b, the single electrons in NO and R-S radicals are not visible. Also, I wonder whether HNO2 is correctly included in the oxidative pathway.

-       Part 3. Enzyme catalyzed regulation of S-nitrosated proteins. In the introduction it may be added that protein nitrosothiols are sometimes reduced by GSH. So, for some proteins at least, GSH works in non-enzymatic denitrosation. Since GSH denitrosation gives rise to GSNO, this reaction may be physiologically connceted to the last part of the review dealing with GSNOR and AKRs.

-       Line 213 and 222: since the resolving Cys of TRXs is mostly buried and usually protonated (line 222), TRXs can hardly contain two protruding redox active Cys (line 213).

-       Line 380, for the sake of clarity it might be reminded that any redoxin has its own reduction system (not only GRX). Also SRX need a reductant, in the form of TRX.

Author Response

We would like to thank both experts for their comments on our work. The manuscript in the current version was modified according to the suggestions of the reviewers, and we hope it increased the quality of our work.

Our responses are marked in red for easier evaluation.

This review by Patrick Treffon and Elizabeth Vierling provides an interesting overview of the biochemistry of S-nitrosation and S-denitrosation of proteins in plants, with emphasis on thioredoxins h and aldo-keto reductases for their direct and indirect potential role in protein S-denitrosation. The review is well written and organized. I have only a short list of minor comments:

-       Line 53: I wouldn’t say that “Redox-active regulatory thiols are conserved in protein catalytic sites” without adding the information that redox-active regulatory thiols are quite often away from proteins active sites and yet responsible of the regulation of their activity.

Response: We changed the sentence to: Redox-active regulatory thiols can be found as conserved residues in protein catalytic sites or in regions distinct from the active site and participate in several mechanisms like thiol-disulfide exchange reactions, for example in thioredoxins (TRXs), glutaredoxins (GRXs) and peroxiredoxins (PRXs), or in electron transfer reactions in the enzymes glutathione reductase (GR) and thioredoxin reductase (NTR) [21,22].

-       Line 76: As far as I know, sulfiredoxins were shown to reduce sulfinic to sulfenic acids in peroxiredoxins only. It seems more an exception rather than a rule. The statement instead seems to suggest that TRXs, GRXs and SRX are equally effective.

Response: Sentence has been changed to: “While oxidation states up to R-SO2H can be reduced enzymatically by TRXs and GRXs, no reduction system for overoxidized Cys (R-SO3H) has been identified [28–30]. Besides the TRX and GRX system, sulfiredoxins (SRX) have been identified as specific enzymes catalyzing the reduction of overoxidized (R-SO2H) peroxiredoxins.

-       Figure 1a might be improved. Some suggestions: GSSG and GSNO should be close to the arrow from P-S- to P-SSG (not the opposite one). H2O2 should be added above the arrow from P-Sto P-SOH. Maybe another arrow should be added to show that disulfide bonds may derive from thiolates, not only from sulfenic acids. In Figure 1b, the single electrons in NO and R-S radicals are not visible. Also, I wonder whether HNO2 is correctly included in the oxidative pathway.

Response: Figure 1 has been changed accordingly.

-       Part 3. Enzyme catalyzed regulation of S-nitrosated proteins. In the introduction it may be added that protein nitrosothiols are sometimes reduced by GSH. So, for some proteins at least, GSH works in non-enzymatic denitrosation. Since GSH denitrosation gives rise to GSNO, this reaction may be physiologically connceted to the last part of the review dealing with GSNOR and AKRs.

Response: The focus of this review article is on the enzymatic denitrosation reactions and the proteins that are involved. Others already published on the role of GSH in this process, and we therefore added the following to the manuscript (line 204):

In addition, GSH has been reported to effect the non-enzymatic S-denitrosation reaction of target proteins (reviewed in [77]), such as Arabidopsis GSNOR and GAPDH [78,79].

-       Line 213 and 222: since the resolving Cys of TRXs is mostly buried and usually protonated (line 222), TRXs can hardly contain two protruding redox active Cys (line 213).

Response: Changed the sentence in line 214 to: “TRXs are ubiquitous, multifunctional thiol-disulfide oxidoreductases containing two Cys residues in a conserved active site motif (WC[G/P]PC) [48]

-       Line 380, for the sake of clarity it might be reminded that any redoxin has its own reduction system (not only GRX). Also SRX need a reductant, in the form of TRX.

Response:  Sentence has been changed to: “However, in contrast to the TRX system which uses NADPH and NTR, GRXs are dependent on GSH and glutathione reductase for their activity [28].”